# Health-Related Quality of Life (HRQoL) in Sarcoidosis: Diagnosis, Management, and Health Outcomes

**DOI:** 10.3390/diagnostics11061089

**Published:** 2021-06-15

**Authors:** Lesley Ann Saketkoo, Anne-Marie Russell, Kelly Jensen, Jessica Mandizha, Jinny Tavee, Jacqui Newton, Frank Rivera, Mike Howie, Rodney Reese, Melanie Goodman, Patricia Hart, Bert Strookappe, Jolanda De Vries, Misha Rosenbach, Mary Beth Scholand, Mathew R. Lammi, Marjon Elfferich, Elyse Lower, Robert P. Baughman, Nadera Sweiss, Marc A. Judson, Marjolein Drent

**Affiliations:** 1New Orleans Scleroderma and Sarcoidosis Patient Care and Research Center, New Orleans, LA 70112, USA; kjensen1@tulane.edu (K.J.); mlammi@lsuhsc.edu (M.R.L.); 2Comprehensive Pulmonary Hypertension Center and Interstitial Lung Disease Clinic Programs, University Medical Center, New Orleans, LA 70112, USA; 3Section of Pulmonary Medicine, Louisiana State University School of Medicine, New Orleans, LA 70112, USA; 4Tulane University School of Medicine, Tulane University, New Orleans, LA 70112, USA; 5College of Medicine and Health, University of Exeter, Devon EX1 2LU, UK; 6Imperial College Healthcare NHS Foundation Trust, London W2 1NY, UK; 7Department of Internal Medicine, Oregon Health and Science University, Portland, OR 97239, USA; 8Respiratory Medicine, Royal Devon and Exeter Hospital NHS Foundation Trust, Exeter EX2 5DW, UK; jessica.mandizha@nhs.net; 9Department of Neurology, National Jewish Health, Denver, CO 80206, USA; taveej@njhealth.org; 10Sarcoidosis UK, China Works, Black Prince Road, London SE1 7SJ, UK; jacquijnewton@gmail.com (J.N.); mikesarco73@gmail.com (M.H.); 11Foundation for Sarcoidosis Research, Chicago, IL 60614, USA; fjr311@gmail.com (F.R.); rreese1956@gmail.com (R.R.); 12National Sarcoidosis Support Group, Stronger than Sarcoidosis, New York, NY 11727, USA; 13CGI UK, Space Defense & Intelligence (Cyber Security Operations), London EC3M 3BY, UK; 14Sarcoidosis Awareness Foundation of Louisiana, Baton Rouge, LA 70812, USA; 15New Orleans Sarcoidosis Support Group, New Orleans, LA 70112, USA; nosarcoidosis@gmail.com; 16iHart Wellness Holistic Approach to Sarcoidosis Certified Health & Wellness Coach, International Association of Professionals, New York, NY 11727, USA; pebhart@gmail.com; 17Department of Physiotherapy, Gelderse Vallei Hospital, 10, 6716 RP Ede, The Netherlands; strookappeb@zgv.nl (B.S.); marjon.elfferich@hetnet.nl (M.E.); 18ildcare Foundation Research Team, 6711 NR Ede, The Netherlands; m.drent@hetnet.nl (M.D.); 19Admiraal de Ruyter Hospital (Adrz), 114, 4462 RA Goes, The Netherlands; jolanda.devries@adrz.nl; 20Department of Medical and Clinical Psychology, Tilburg University, 5037 AB Tilburg, The Netherlands; 21Cutaneous Sarcoidosis Clinic, Department of Dermatology, University of Pennsylvania, Philadelphia, PA 19104, USA; Misha.Rosenbach@pennmedicine.upenn.edu; 22Division of Pulmonary Medicine, Interstitial Lung Disease Center, University of Utah, Salt Lake City, UT 84132, USA; scholand@genetics.utah.edu; 23Department of Medicine, University of Cincinnati Medical Center, Cincinnati, OH 45267, USA; lowere@ucmail.uc.edu (E.L.); BAUGHMRP@ucmail.uc.edu (R.P.B.); 24Division of Rheumatology, Department of Medicine, University of Illinois at Chicago, Chicago, IL 60612, USA; nsweiss@uic.edu; 25Division of Pulmonary Medicine and Critical Care, Albany Medical College, Albany, NY 12208, USA; JudsonM@amc.edu; 26Interstitial Lung Diseases (ILD) Center of Excellence, Department of Pulmonology, St. Antonius Hospital, Koekoekslaan 1, 3435 CM Nieuwegein, The Netherlands; 27Department of Pharmacology and Toxicology, Faculty of Health and Life Sciences, Maastricht University, 40, 6229 ER Maastricht, The Netherlands

**Keywords:** sarcoidosis, quality of life, symptom burden, symptom distress, shared decision making, patient-centeredness, patient centered care, mindfulness, exercise, physical activity

## Abstract

Health-related quality of life (HRQoL), though rarely considered as a primary endpoint in clinical trials, may be the single outcome reflective of patient priorities when living with a health condition. HRQoL is a multi-dimensional concept that reflects the degree to which a health condition interferes with participation in and fulfillment of important life areas. HRQoL is intended to capture the composite degree of physical, physiologic, psychological, and social impairment resulting from symptom burden, patient-perceived disease severity, and treatment side effects. Diminished HRQoL expectedly correlates to worsening disability and death; but interventions addressing HRQoL are linked to *increased survival*. Sarcoidosis, being a multi-organ system disease, is associated with a diffuse array of manifestations resulting in multiple symptoms, complications, and medication-related side effects that are linked to reduced HRQoL. Diminished HRQoL in sarcoidosis is related to decreased physical function, pain, significant loss of income, absence from work, and strain on personal relationships. Symptom distress can result clearly from a sarcoidosis manifestation (e.g., ocular pain, breathlessness, cough) but may also be non-specific, such as pain or fatigue. More complex, a single non-specific symptom, e.g., fatigue may be directly sarcoidosis-derived (e.g., inflammatory state, neurologic, hormonal, cardiopulmonary), medication-related (e.g., anemia, sleeplessness, weight gain, sub-clinical infection), or an indirect complication (e.g., sleep apnea, physical deconditioning, depression). Identifying and distinguishing underlying causes of impaired HRQoL provides opportunity for treatment strategies that can greatly impact a patient’s function, well-being, and disease outcomes. Herein, we present a reference manual that describes the current state of knowledge in sarcoidosis-related HRQoL and distinguish between diverse causes of symptom distress and other influences on sarcoidosis-related HRQoL. We provide tools to assess, investigate, and diagnose compromised HRQoL and its influencers. Strategies to address modifiable HRQoL factors through palliation of symptoms and methods to improve the sarcoidosis health profile are outlined; as well as a proposed research agenda in sarcoidosis-related HRQoL.

## 1. Introduction

Sarcoidosis is a phenotypically heterogeneous, systemic disease of unknown etiology characterized pathologically by the presence of non-caseating granulomas in one or multiple organs. The presentation of sarcoidosis is highly variable. Though sarcoidosis potentially resides in any organ and most commonly recognized with pulmonary, cutaneous, ophthalmologic involvement; neurologic, cardiac, gastrointestinal, hepatic, and renal involvement are likely under-recognized. The presence of granulomas may be silent or cause severe or life-threatening organ dysfunction resulting in multiple and diverse symptoms that impair physical function and psycho-social realms of function due to direct effects of disease or treatment effects. Sarcoidosis-related impairments can impede routine activities of daily living (ADLs) and disrupt critical life areas: work, family, and social/leisure; and also impact psychological well-being.

When relying predominantly on positive biomarkers, radiologic, and physiologic testing to guide treatment, a patient’s physical, psychological, and cognitive impairment in sarcoidosis can often be overlooked by clinicians [1,2]. Clinician-recognition that “absence of evidence does not mean evidence of absence”, especially when PET/CT is not available, is crucial in considering patient history. It should be acknowledged that sarcoidosis patients prioritize quality of life issues over most objective clinical tests to assess their disease [1,2].

Health-related quality of life (HRQoL), though rarely a primary endpoint in clinical trials, may be the single outcome most reflective of patient priorities when living with a health condition [3]. HRQoL is a multi-dimensional concept that reflects the degree to which one’s health condition impairs the ease with which one is able to interface with crucial areas of life activity: making a living, engaging with loved ones, and pursuing life’s interests [4]. Health status, or physical function, is only one component of HRQoL, and therefore measures that are specific to health status/physical function do not measure the extent of HRQoL [5]. HRQoL reflects how important life areas are enhanced or diminished by the health condition, treatment, and environmental influences. Adopting, cultivating, and advocating for patient-centered approaches throughout all aspects of healthcare delivery, naturally address and augment HRQoL as experienced by the patient and their family [6]. HRQoL in sarcoidosis is an expansive topic; this chapter is meant to serve as an abbreviated ‘reference manual’ to cultivate familiarity with HRQoL aspects amenable to intervention, develop an approach to assessing HRQoL, and consider a preliminary research agenda in sarcoidosis-related HRQoL.

## 2. Health-Related Quality of Life (HRQoL)

The evolution of health paradigms has evolved to recognize that physical, social and psychological health are inextricably inter-influential. This is outlined as the Bio–Psycho–Social Model of Health (Figure 1) [7]. HRQoL captures the composite impact of symptom burden, patient-perceived disease severity, treatment side effects, and healthcare interactions on physical, psychological, and social well-being as well as the degree of ease with which one is able to participate in important life areas (family, work/school, social, hobbies).

Though increasingly recognized for its importance of what is meaningful to patients, it has remained problematic to incorporate HRQoL issues seamlessly within the clinician-centered medical management paradigm [1,8,9]. The complexity of interpreting multiple domains that comprise HRQoL, e.g., physical, social, mental, cognitive, and spiritual, alongside so-called standard ‘objective’ measures of health status, largely remain a challenge for western medicine, to find a practical actionable approach. Further, ‘objective’ disease assessments may not correlate with reported symptomatology nor impaired HRQoL.

HRQoL can be favorably or negatively (Figure 2) influenced by [10]:*Personal factors* include a patient’s intrinsic potential for adaptability and coping behavior, length of time living with a condition, increasing familiarity with self-management strategies for symptoms and impairment;*Environmental factors* include the extent of family support, financial resources, as well as assistive aids, devices, or techniques that improve physical, mental, or emotional function. This includes access to care and expressly, in the case of sarcoidosis, access to clinicians with sufficient knowledge of sarcoidosis [8];*Symptom burden* includes disease-related symptoms and impairment, medication side effects [11], and the psychological impact of living with the health condition.

Any of these areas can be influenced by clinician-initiated interventions; the need for which generally increases in complex diseases, such as sarcoidosis [1,8,9].

### 2.1. Symptom and Impairment Burden

Symptom distress has been shown to have an inverse relationship with both HRQoL and survival (Figure 2). This relationship has been well-studied in numerous health conditions, demonstrating that lower physical function and HRQoL may be important predictors of worse health independently of traditional objective disease severity measures [12,13,14,15,16,17]. A number of life-threatening conditions have demonstrated that HRQoL is predictive of mortality independent of objective severity parameters, such as tumor size, vascular events, or organ damage scores; and higher HRQoL scores are associated with survival [14,15,17]. Across oncological diseases, symptom distress, predictably, was an impediment to HRQoL and survival. While interventions to decrease symptoms and symptom distress appeared to extend survival [18,19,20,21]. The clinician cognizant of the diverse potential impairments in any health condition is in a better position to increase a patient’s HRQoL status.

However, the clinician who is current on the expanse and complexity of sarcoidosis care, is also better able to gauge the likelihood from where symptoms arise along the *disease activity < --- > damage trajectory* (Figure 3), leading to prompt recognition, diagnosis, and prevention of the HRQoL diminishing health occurrences reported by patients [1,8,9,11]. These include misdiagnosis, missed opportunities for treatment, avoidance of common disease, and medication related complications, e.g., hypercalcemia/vitamin D over-supplementation, sun protection, etc. This also encompasses pharmacological and non-pharmacological palliative measures warranted to overcome the confines of irreversible damage, such as diuresis for heart failure in non-active cardiac sarcoidosis or enlisting physical, respiratory, or occupational therapy to augment function in disability resultant damage from non-active neurosarcoidosis or pulmonary fibrosis [1,8,9,11].

Being a heterogeneous multi-organ system disease, sarcoidosis, is associated with a diffuse array of both reversible (still treatable and unlikely to leave damage with timely treatment) and irreversible (pharmacologically untreatable damage) manifestations. Sarcoidosis may cause symptoms that are organ- and non-organ-related, along with less easily definable or non-specific symptoms, such as fatigue, exercise intolerance, breathlessness, and depressive symptoms, which are of the most frequent physical symptoms experienced by people living with sarcoidosis [22,23]. Critical thinking that considers this wide variation in disease phenotype, symptomatology, and disease behavior, alongside the inverse relationship between treatable inflammation and irreversible fibrotic damage (Figure 3), lends itself to interventions that potentially diminish these HRQoL struggles.

The type, number, and severity of physical symptoms are widely variable in sarcoidosis due to the multi-organ system nature of the disease and the various treatments (Table 1), and all cases warrant careful consideration of both pharmacological and non-pharmacological interventions. For each of these factors, treatment approaches hinge upon whether the symptoms are the result of (Figure 3):*Sarcoidosis disease activity* clinically suggests granulomatous inflammation impacting health status and, therefore, impairment is potentially reversible with quiescence of disease activity, either through pharmacological treatment or disease self-remission.*Sarcoidosis damage* is the damage and scarring left in the wake of prior destructive sarcoidosis inflammation and granulomatous activity; this tissue damage is not irreversible.A *combination* or an *overlap* of active disease and irreversible damage; this chronic progressive phenotype suggests reversibility of impairment depends on extent of damage versus treatable inflammatory disease.*Treatment-related* effects, complications, or adverse events, e.g., amiodarone thyroiditis resulting from arrhythmia treatment, electrolyte disturbance from diuresis from cardiomyopathy treatment, or steroid-related myopathy, or weight gain, etc. [11,24].*Pre-existing and co-existing comorbidity* can confound symptom interpretations and always deserve differential diagnostic attention, e.g., pre-existing hyperthyroidism with presentation with tachycardia in otherwise controlled sarcoidosis, or recent lymphoma mimicking sarcoidosis-like symptoms, even with concomitant granulomatous involvement.*Non-sarcoidosis related*, e.g., coronary artery involvement, etc.

In summary, consideration of symptoms in sarcoidosis requires (a) robust critical thinking, and (b) always keeping in mind that though symptoms that are perceived as mild from a clinical perspective, may be of significant distress to patients in order to have a beneficial impact on HRQoL.

### 2.2. Participation

Participation is the ease with which a person is able to interface with important life fulfillment areas such as work/education, family, social and leisure. Symptoms distress, whether physical, non-specific, or psychological can impede to varying degrees. Improvements in symptom distress and reduction of impairment can lessen this struggle and hopefully restoring ease at work and family. When symptoms/impairment are not modifiable with treatment or assistive devices, then expectations of in that life area, e.g., work or household, may need temporary or more long-term adjustment.

#### 2.2.1. Work Life

Work ability/productivity is a composite concept of presenteeism (being present for work, versus absenteeism) and performance and provides a robust example of bio-psycho-social convergence in sarcoidosis [25]. Several recent studies examined the societal burden of sarcoidosis in terms of work ability and financial cost with average sick days reported as much higher than controls at 30 days yearly, which continues to be much higher than general population at least five years out from diagnosis [26,27,28,29]. Sarcoidosis-related absenteeism in one study revealed an averaged 8% income lost yearly [26,27,28,29]. These are compounded by unforeseen disease-related financial losses including absenteeism of family for appointments/hospitalizations, travel, medication/healthcare co-pays, and additional costs related to symptom amelioration not covered by insurance (over-the-counter medications, massage, acupuncture, adaptive devices and home/work modifications). These financial hardships are experienced more severely by those already economically disadvantaged [30,31,32].

Impairment of work productivity or satisfaction adversely impacts several aspects of HRQoL in terms of financial stability, self-worth, family hardship, and workplace community. The degree to which such losses interfere with these dynamics are in turn influenced by these dynamic’s intrinsic qualities, such as economic status and availability of disposable income/savings, workplace policies, and co-worker attitudes, hierarchical position, supportive relationships, and even weather [33].

Tackling these issues requires the careful development and systematic implementations of education for patients, families, and employers that include anticipatory guidance on preventive health measures, attention to potential financial changes, solution-driven workplace philosophies, such as work-from-home and sick-day-donation programs; as well as re-structuring of clinical throughputs, such that clinic and diagnostic appointments are consolidated on single visits to reduce absenteeism, co-pay, and travel costs, along with more robust efforts to develop home therapeutic and health monitoring programs [34,35].

Assessing sarcoidosis-related impact on work ability and quantification of disease burden for disability claims may also impact HRQoL. Sarcoidosis patients may experience disease severity and disability beyond the assessment protocols and measures used to assess disability claims [25], e.g., lung function appears to be preserved, yet fatigue and cognitive failure present strong limitations on work productivity. Lung function test results should not be considered the only reliable criterion for disability assessment. However, since the COVID-19 pandemic, the medical and employment worlds are being forced to recognize and accommodate the potential long-term debilitation as a reality also in other chronic conditions, such as sarcoidosis [8].

#### 2.2.2. Family

As information on HRQoL in sarcoidosis becomes more available, characterization of family and social impact are likely to emerge. Extrapolating from available investigations of family and social impact in other health conditions is reasonable. Symptom burden (pain, fatigue, dyspnea, nausea, depression, anxiety) often interferes with ability to connect with and engage in pleasure, responsibilities and easy communication with our loved ones.

While central nervous system (CNS) involvement is recognized to interfere with neuro-hormonal aspects of sexual activity, no studies have assessed the impact of sarcoidosis symptom burden on intimacy. However, fatigue, dyspnea, pain, physical deconditioning, and psychological effects of disease, as extrapolated from other diseases [36], are hypothesized to diminish HRQoL as it relates to seeking and maintaining intimate relationships in sarcoidosis.

#### 2.2.3. Social Life

In terms of social engagement, symptoms such as fatigue, pain, dyspnea, cough, and mobility impairment can create a sense of self-consciousness and isolation [37,38]. Further, in sarcoidosis [39] and other conditions [37,38], lack of public awareness, and sometimes stigma, fuel misunderstanding and isolation. Patients with significant non-specific and ‘invisible’ impairment internalize cultural stigmas related to ‘laziness’ and ‘will-power’, resulting in a downward complex interplay of physical and psychological symptomatology concretizing isolation [2,40].

## 3. Patient-Centeredness

Clinician-initiated strategies to increase HRQoL begin with an environment whereby a patient feels respected, listened to, and believed, followed by open-minded investigations based on patient history. Subsequent efforts target the patient’s report of symptom burden, physical impairment, as well as potential adaptations to home/work participation. This occurs in the context of continued thoughtful consideration of resources at hand, such as other available programs and inter-disciplinary care members, such as occupational or physical therapy.

Patient-centeredness is at the heart of approaches and actions directed toward increasing a patient’s HRQoL within the sphere of healthcare delivery. Patient-centeredness broadly translates to patients’ voiced priorities, concerns, and perspectives that can guide anything to do with care on large-scale or individual levels. This includes treatment/management, communication, clinical environment, policy and procedures, in addition to delivery of and access to care. Figure 2 outlines the powerful impact that clinician-related factors have on HRQoL.

Patient-centeredness took hold in the 1960s with patients being viewed as unique human beings rather than cohorts of diseased patients and biological processes being of partial relevance to the health equation, namely the psychosocial context with HRQoL the concept to be measured [41]. Further, the expertise of patients and their evolution into a decision-making partner, in clinical, research and policymaking arenas, is gaining increasing value, as evidenced by large shifts in national healthcare policies [42].

However, in sarcoidosis, very scarce literature describes patient perspective of disease, treatment, HRQoL and the extensive difficulties of navigating and being heard within the healthcare systems, which is largely perceived lacking knowledge of current sarcoidosis care [8,9]. Eight inter-related principles essential to patient-centered care [43] are defined by The Picker Institute and integrated into the NHS Patient Experience Framework. (Figure 4).
Respect for patients’ values, preferences, and expressed needs;Coordination and integration of care;Information, communication, and education;Physical comfort;Emotional support and alleviation of fear and anxiety;Involvement of family and friends;Continuity and transition;Access to care.

Additionally, attention to cultural and spiritual differences in care discussions may be a key patient-centered factor in health outcomes [8,44].

### 3.1. Patient-Centered Environments

The baseline stress that comes with serious illnesses significantly impacts health outcomes. Beyond this stress, patient/family stress is compounded by a myriad of logistical and financial burdens, such as transportation, work absenteeism for patients and family members, appointment conflicts, and health insurance challenges. Telehealth, where appropriate, for home-based or work-based appointments and integration of home-based testing, e.g., spirometry, oximetry, electrocardiogram etc. [34,35], can offset multi-factorial burdens on patients/family members.

The clinical environment can either perpetuate further stress or be a source of reassurance and *stress-reduction* for the patient/family. Noise, complicated navigation between service points, difficult parking/transportation, over-crowding, unpleasant odors, and harsh or too dim lighting, are environmental stressors on patients and their families. Patient-centered environments support ease of navigation, appointments, procedures, and provide safe, physically comfortable, and welcoming atmospheres with natural light, green spaces, and vegetal life where possible, as well as floral, herbal, or woody natural fragrances [45,46]. Patient-centered environments make people feel happy and repeatedly demonstrate benefits on health outcomes [47]. These outcomes include shorter hospitalizations, reduced post-surgical complications, and the need for pain medications; improved surgical outcomes, depression, and cognition; as well as increased hospital revenue and employee productivity [47,48]. Large structural changes are not required to be effective; simple beautification and logistical strategies, such as clear sign-posting, friendly and helpful staff, etc., make big differences to patient/family perceptions of care.

Stress-relieving education (see section below) and the support of integrative approaches (e.g., meditation, healthy diet, exercise, tai chi, yoga, singing) [49,50,51] in the clinical environment through on-site programs or apprising patients of integrated opportunities for stress-reduction set a caring tone. Supporting these concepts as a routine part of the clinical environment, result in favorable health outcomes and improve staff well-being.

### 3.2. Communication

Regardless of the type or severity of an HRQoL factor, clinician communicated recognition of patient/family suffering and potential isolation, can provide the support patients/family need to initiate self-motivated modifications [52,53]. Thus, small simple gestures of communications can have a powerful impact on patients/families.

#### 3.2.1. Trauma-Informed Patient Communication

Professional societies and initiatives are promoting the adoption of trauma-informed/sensitive practices in all patient-care [54]. This promotion acknowledges that trauma—whether related to sexual, racial, or the trauma of having a rare complex disease—exists [55,56], is pervasive, and is predominantly not recognizable. It also acknowledges that the propensity for inadvertent medicalized re-traumatization is common and, therefore, trauma-informed patient care [57] should prevail as a baseline communication strategy. Trauma-informed care is simply a commitment to respect, compassion, safety, and patient empowerment [58]. It characterizes a patient-centered relationship built upon compassionate principles as guided by the six simple trauma-informed practices [52,53,59]:Safety;Trustworthiness and transparency;Peer support;Collaboration and mutuality;Empowerment and choice;Cultural, historical, and gender considerations.

Medical training supports these concepts; however, upholding these principles are vulnerable to the rapid erosion into unconscious bias as time- and revenue-based pressures compete with authentic humanistic communication. The presence of a respectful, compassionate clinical relationship is a powerful alliance that can help lift a veil defeatism and reveal potential and hopefulness [54]. This strength of compassionate communication appears to be true in the context of acknowledged poor prognoses [52,53].

#### 3.2.2. Shared Decision-Making

Shared decision-making (SDM) is one of the most integral practices to patient-centeredness. Sarcoidosis, being a complex disease in terms of manifestations, symptomatology, and coincident medications, make SDM more demanding on the patient–clinician dyad. In shared decision-making, clinicians share in the process, but final decisions belong to the patient.

The role of the clinician is to convey, as much as possible, all the elements that are known and unknown, and to assist the patient in navigating through the information that will support them in making the best decisions according to their priorities and preferences. The respectful support of the clinician in this process is tied to patient well-being [39,60], confidence in and adherence to treatment, and improved health outcomes [39].

Shared decision-making assesses risks of testing and treatments weighed against the benefits in terms of patients’ long- and short-term health and wellbeing. Assessment of patient expectations before initiation of treatment, followed by systematic baseline and interval evaluation of how a patient is experiencing the disease, enhance shared-decision making and patient adherence.

Specifically in sarcoidosis, treatment expectations will hinge on formulaic discussions weighing:*symptomatology/impairment resulting from disease activity versus residual disease damage**+ risk of serious organ dysfunction versus low or no progression or chance of remission**+ disease monitoring versus active treatment**+ anticipated benefit of treatment versus risk of toxicity**= preliminary decision*

SDM is informed by patient priorities and preferences at that time and, therefore, shared decision-making is a dynamic process overtime. Disease behavior and therapeutic responsiveness and toxicity, along with life circumstances, may shift previously stated priorities and preferences; and as part of the shared decision-making construct, clinicians remain open and responsive to this.

Sarcoidosis skin involvement provides examples reverberant for other areas of sarcoidosis organ involvement of how SDM is pivotal in bridging clinician–patient perception to impact HRQoL:Cutaneous sarcoidosis, as an example, could lead to a provider-patient mismatch in terms of disease impact and treatment. In a case of non-facial cutaneous involvement with no other organ involvement necessitating treatment, discussion allows the patient to convey the presence and/or burden of physical symptoms (e.g., pruritus, burning, etc.), and also any psychological impact these symptoms and the cosmetic appearance might have. After the clinician explains treatment options with their benefit vs risk implications (e.g., hydroxychloroquine, topical steroids, etc.), if all symptom factors are none to minimal, the patient may decide to delay treatment in favor of observation; while the patient may opt for treatment if the cosmetic appearance or physical symptom burden is noticeable to them, thus outweighing medication-related side effects or costs.In a case of lupus pernio, aggressive treatment is often necessary to achieve adequate disease control (e.g., use of biologic therapy). As patients with this phenotype are often quite impacted by the disease, and typically have extra-cutaneous disease requiring treatment, generally the patient and clinician are aligned in the necessity of treating active disease. However, once active disease has been treated, the clinician may see a lack of active granulomatous inflammation as a satisfactory end result, whereas residual damage in the form of dyspigmentation and scarring may be quite troubling to patients cosmetically and psychosocially. Without discussion, the clinician may fail to recognize that the scarring is causing the patient significant distress (e.g., depressive symptoms, self-esteem). With discussion, the clinician learns of the patient’s distress, and is now in a position to provide support via communicating recognition of the patient’s distress, psychological support, and considering possible cosmetic interventions (Box 1).

Box 1Checklist to support shared decision-making *(Courtesy of LA Saketkoo, rights reserved).*Shared Decision-Making Checklist
❏**Name the patient’s items of concern** as presented by the patient and if possible, which are highest priority❏**Ascertain patient’s thoughts** on the potential underlying cause/s❏**Name the items of concern** from the clinical perspective including short and long-term (e.g. potential progressive damage, associated abrupt complications etc)❏**Respond to patient’s perceptions** of potential cause in support of & clarifying divergence from patient perceptions. Remain transparent in what is known, unknown, yet to be known and that which requires researching by the clinician❏**Name the treatment options available**, including any nonpharmacological with particular attention those suggested by the patient❏**Discuss safety, side effects and efficacy** (including anticipated onset) of available therapies and those suggested by the patient❏
**Assess Patient Expectations of treatment**
❏**Set Treatment Expectations** including prognosis, anticipated degree of symptom/impairments resolution, cure versus slowing progression, disease activity versus damage


### 3.3. Family as an Extension of the Patient

Sarcoidosis impacts the family members and/or those who provide an informal caring or supportive role. Caregiver health and well-being directly impacts patient health outcomes [6]. Family members instinctively know their care influences their loved one’s health outcomes [61,62,63]. Family responsibilities can extend beyond assisting with activities of daily living (ADLs), nutrition, organizing and administering medications, managing finances, and dealing with the healthcare system challenges, often while living with anticipatory grief [61,62,63,64].

Greater than 90% of caregivers report declining health, depression, with caregiving taking a toll on daily activities, while being plagued emotionally by worry, often about the future, and fearful of their loved one’s suffering and death. Sleep loss occurred in 67% because of worrying, helping with personal hygiene or medication, or the need to check on their loved one. Greater than 50% reported caregiving impacting family relationships, work and study, and finances.

These expressions of distress translate to detrimental health outcomes for caregivers and patients [65,66,67]. Iterative studies identify higher caregiver strain as leading to increased depression, stress and increased mortality. Increased mortality at 63% generally, and at 84% at the highest levels of strain, with 134% increase for cardiovascular events in caregivers over a mere 30-month period. Consistently, studies demonstrate increased caregiver depression as proportional to caregiver strain and increasing cardiovascular mortality.

Depressive symptoms, the perception of being overwhelmed, and poor scores on caregiver scales predict poor caregiver health outcomes. Decreased strain, depression, and adverse outcomes result from targeted interventions, including stress-reduction strategies, education on caregiver strain, anticipatory stress, related depression, as well as empathetic communication [68], providing caregiver with available self-care resources. *Advocacy* for healthcare policy recognizes the financial and physical demands of caring for a loved one, such that practical frameworks might be instituted to support them [69,70].

### 3.4. Patient Advocacy Organizations

Patient organizations have an important role in education, research, and advocacy. Patient organizations raise awareness about health conditions in the lay and medical communities and targeting education to these populations to impact health outcomes including HRQoL [8]. Patient organizations often are sensitized to and provide educational and other support for situations unique to the disease, disease-related transitions, and areas of greatest concern to patients. Patient organizations, networks, and support groups have the lens of lived experiences that facilitate personal bonds and shared-information on living that are beyond the abilities of a healthcare team. Patients and their loved ones should be apprised of patient organizations, which can be an integral part of patient and family sustenance.

## 4. Patient-Centered and HRQoL Instruments

A range of instruments can support the patient, clinician, and researcher in patient-centered processes, some of which we touch upon below. These include:*Patient-reported outcome measures* (PROMs) that capture changes in symptoms, health status perception, or HRQoL;*Patient-reported experience measures* (PREMs) that identify patient-experienced strengths and weakness of healthcare delivery systems in order to improve the system and thereby, patient experience and HRQoL;*Patient engagement* (or *activation*) *measures* (PEMs/PAMs) that assess areas such as disease or medication knowledge, health systems familiarity, or lifestyle awareness, so that patients might receive education or counseling to strengthen self-management skills, and thereby enhance HRQoL;*Patient preference measures* supporting patient decision-making or provide data for value-based health economic choices;*Clinical checklists* that enhance patient outcomes, through decreasing complications and supporting patient-centered care, all of which increase HRQoL.

### 4.1. Overview of Assessments in Sarcoidosis

Registries, government, and commercial databases help to estimate crude aspects of disease burden. However, patient-reported measures can provide finer information on both group and individualized levels for epidemiological, procedural, as well as clinical assessment.

PROMs, in particular, are instruments comprised of questions answered by patients either in clinical care or clinical trial settings to (usually) assess symptom burden (e.g., dyspnea, cough, fatigue, or cognition), physical function, health status, or the more expansive construct of HRQoL. PROMs are considered to be either ‘generic’ or ‘disease-specific’. Disease-specific measures are those that have been developed to capture information uniquely relevant to a particular health condition.

Generic measures are used and validated across a wide variety of health conditions and even general or healthy populations. Commonly used generic PROMs include the SF-36, EQ-5D, or PROMIS. Generic HRQoL measures can be used alone when disease-specific measures are not yet available, or alongside disease-specific measures to provide construct validity. Generic measures are also used to compare HRQoL and disease burden between and across various diseases, for example, a comparison of HRQoL between lung cancer and interstitial lung disease (ILD), which can guide allocation of resources and policy or program developments.

Disease-specific measures in sarcoidosis include the Sarcoidosis Health Questionnaire (SHQ) [71], King’s Sarcoidosis Questionnaire (KSQ) [72,73,74], and the Sarcoidosis Assessment Tool (SAT) [75]. The Fatigue Assessment Scale (FAS) has been repeatedly validated in sarcoidosis, as well as several other health conditions, to assess fatigue. The FAS, KSQ, and SAT all have defined minimal clinically important difference (MCID), making them useful tools in assessing response to disease and interventions [72,75,76]. The KSQ General Health was found to have an MCID of 8, the KSQ Lung Scale of 4, and the Patient Global Assessment Scale of 2, each of these values capturing >90% of the parameters studied. These MCID values also discriminated between changes in other HRQoL instruments. Further assessment tools are discussed below under specific symptoms.

For clinical practice and as an exploratory endpoint in diseases with heterogeneous manifestations, measures, such as Patient Specific Functional Scale (PSFS) [77] and the McMaster Toronto Arthritis patient preference questionnaire (MACTAR) [78], are created individually by patients listing and prioritizing routine concepts most important to them (e.g., symptom, an activity, or task) to be tracked over time. These are then rated on a pre-determined response scale e.g., VAS or Likert, with careful attention to time reference and wording of response scale. An example of *coughing* being the most troubling symptom, the item might become: ‘*In the past week, the degree to which cough limits you*, 0 being ‘*not at all*’ and 10 being ‘*completely unable to function*’. Another example, might be ‘*In the past week, how able I am to read aloud to my children*’, 0 being ‘*no problem reading aloud*’ and 10 being ‘*unable to read aloud at all*’.

### 4.2. Patient-Centered Checklists

Checklists provide clinician support with organized frameworks to improve patient comfort, safety, ability and health outcomes while reducing patient suffering through comprehensive assessment and quality assurance measures [79,80,81]. Further, the quality resulting from comprehensiveness inquiry and management of checklists, fortifies patient/family confidence in their care, strengthening the clinician–patient relationship and thus impacts several HRQoL areas, and reducing clinician burn-out [79,80,81,82]. Checklists might be discrete, work in concert with each other, or coalesce to form a master checklist.

A symptom checklist might include disease- or medication-related symptom queries that might spark investigatory or management considerations, e.g., cough, dyspnea, nausea or diarrhea, or extent of physical activity. While a medication checklist might remind the practitioner that glucocorticoids requires screening for co-morbidities, contra-indications, and/or protective measures for, e.g., bone or gastric. Checklists may prompt screening, investigation, or intervention for, e.g., depression or for sleep apnea. The prevention section of a checklist may include vaccination requirements or update of testing, e.g., annual hepatitis B for certain immunosuppressants. Checklists can also become metrics marking incidence, duration, and change over time. Many examples of checklists exist or can be devised and revised over time to improve quality care.

There are two disciplines related to checklists that can have high impact on HRQoL and help clinicians strengthen sarcoidosis diagnostic approach and management. The first is the systematic consideration of the enlisting members of *the multi-disciplinary team.* Through a needs-based holistic assessment, relevant subspecialists, therapists, etc., are engaged to assist in improving targeted aspects of HRQoL. These efforts are supported by fluent communication between team members to promote consistency in treatment strategies.

The second being algorithmic checklists that facilitate *critical examination of the diverse array of symptomatic or functional changes in a patient’s health status*. Patients report feelings of frustration, worry and regret where continual and repeated delays in diagnosis are encountered. These feelings are attributed to uninformed decision-making by clinicians, hurried dismissal of symptoms as not being relevant to sarcoidosis or lack of specialist proactivity to coordinate overall sarcoidosis care [8]. Hasty escalation of prednisone by clinicians is a commonly communicated patient experience, in the setting of respiratory changes presumptive of sarcoidosis flare without consideration of other common co-existent entities such as mycotic or mycobacterial infections, pulmonary hypertension or heart failure. At the other extreme, patients report having sarcoidosis complications precipitously dismissed by clinicians as not being sarcoidosis-related, e.g., palpitations as anxiety instead of cardiac sarcoidosis, or headache, and feeling weak in cases of neurosarcoidosis [8]. Patients report erosion of confidence with clinical hastiness, whereas diagnostic errors despite thoughtful investigation are acceptable in a complex disease [8].

### 4.3. Operationalizing Instruments

Most instruments, or tools, extract only snapshot in time from a continuum of a patient’s health condition. How tools or instruments are implemented and interpreted, as well as their limitations, are essential considerations. The intention behind quantifying a patient’s health experience and how this information is to be used should be clear and purposeful to both the clinician and patient. Three overarching areas of meaningful use sustaining patient care and being a source of clinician-support:(a)As a *detection tool* to disclose the need for medical or other intervention in regards to patient health or environment, e.g., depression screening, severity scores, symptom scales, clinician checklists.(b)As *a tracking tool* of symptoms, to mark improvement in patient-designated priorities [77], and other patient-reported measures to be trended over time alongside other scores, e.g., to gauge efficacy of treatment, traditional markers, and historical patient events, e.g., hospitalizations, exacerbations, antibiotic/steroid use; thus, enabling patients and clinicians to identify trends leading up to and to prevent recurrent complications [34,83,84].(c)As a *patient–clinician discussion tool,* whereby results provide opportunities to initiate discussions on potential reasons for score changes by the patient followed by the clinician who can then offer additional perspective or clarify any misunderstandings. Further, the psychological, emotional, physical, and intellectual exhaustion and burnout incurred by dedicated clinicians [79,80,81,82], can be offset, in addition to checklists, by the framework that PROMs and other tools provide to support difficult discussions regarding milestone health changes, e.g., need for lung transplantation assessment.

## 5. Common Causes of Symptom Burden in Sarcoidosis

### 5.1. Psychological Distress

Psychological impact of sarcoidosis is vast and far-reaching [5,85,86], with very high rates of anxiety, depressive symptoms, stress, diminished self-esteem, and isolation [85,87,88,89]. Severity of disease, respiratory symptoms, multi-organ involvement, and disease chronicity correlate with degree of depression. Regardless of actual disease status, patients who perceive their conditions as severe may suffer from anxiety and depressive symptoms, underscoring importance of clinician communication. True to this concept, 75% of sarcoidosis patients voice a need for patient education that includes the psychological impact of sarcoidosis. Although depressive symptoms are less likely to drive HRQoL impairment over physical impairments, such as fatigue [90], depressive symptoms are treatable through psychological support or medication and, therefore, should be addressed in the clinic [85,87,89].

A sarcoidosis patient is often beleaguered with short-term and long-term uncertainty. Fluctuating symptoms, plans changing due to plummeting energy levels, logistical burdens related to tablet intake or managing oxygen equipment, or whether to attend appointments in the face of unpaid leave create daily anxiety-laden short-term uncertainties causing decisions [3,8,37,38]. Whereas, continuous concerns about survival, work ability, severe disability, as well as financial, family, and home security can become goading long-term uncertainties that plague one’s psychological functioning and erode well-being.

Further, patients’ psychological distress intensifies with the painful awareness that their illness causes significant worrying and anticipatory grief in their loved ones [6,39], and can impair these relationships [6,8,91]. Thus, patients often strive to minimize the personal impact of their health conditions to protect loved ones [3], but also to preserve independence and self-identity [3,37].

Other sarcoidosis-related symptoms impact mental health in sarcoidosis, such as cognitive dysfunction, pain, pruritus, and disfigurement from cutaneous manifestations, altered appearance from steroid-related striae, acne, Cushingoid features, or weight gain, as well as decline in physical function. Importantly, yet diagnosed neurosarcoidosis, sarcoidosis-related hormonal dysfunction, and elevation vitamin D/calcium levels require consideration. Further, glucocorticoids, as a primary source or through impaired sleep quality, disrupt mental states with swings from mania to depression, dysphoria, cognitive impairment, and even psychosis.

### 5.2. Cognition

As with fatigue, exercise intolerance, and depressive symptoms, cognitive symptoms related to impaired concentration may persist despite ‘objective’ diagnostic measures of sarcoidosis. Cognitive impairment can manifest as problems with memory, attention, and concentration. Cognitive impairment negatively effects medication adherence and other areas of self-management. The Cognitive Function Questionnaire (CFQ) is a global subjective screening tool for cognitive impairment. More than half of patients with neurosarcoidosis report cognitive deficits, compared to one-third of a general sarcoidosis population [92]. Fatigue and small fiber neuropathy play a role in cognitive failure [92], with the presence of cognitive failure being an independent predictor of fatigue [40]. Interestingly, the only factor associated with a significantly higher level of cognition was recent TNF-alpha-inhibition [93] Again, such findings re-assert the question of sub-radiological and treatable sarcoidosis being present [94]. In addition to optimally suppressing sarcoidosis, psychological interventions that focus on coping, stress reduction, cognitive-behavioral therapy, and mindfulness-based cognitive therapy can support cognitive function.

### 5.3. Pain

Pain in sarcoidosis is the result of diverse causes or combination of causes; requiring a carefully honed approach to distinguish between qualities of symptom experience that include pain type, trajectory, duration, and other factors, such as current health state and current medications to identify most likely causes. Pain often is compounded with patient frustrations of lack of careful enquiry for treatable causes or because of fear or implications of seeking pain medications. Due to lack of practiced acumen or clinic time, pain without an overt and immediately obvious cause, tends to be dismissed as a hopeless endeavor, not related to sarcoidosis, or, as with many with multi-systems, thrown into the default bin of ‘fibromyalgia’, which, by definition, is a diagnosis of exclusion [95,96]. The clinician overseeing the sarcoidosis-related health concerns, regardless of primary specialty, bears the responsibility of initiating investigation to identify cause.

*Small Fiber Neuropathy (SFN)* is most recognized in sarcoidosis for neuropathic pain, paresthesia, allodynia, and temperature sensation dysfunction. SFN-related pain tends to be regional, such as skin involvement. SFN-related chest pain in sarcoidosis, though should be investigated for cardiopulmonary involvement accordingly, occurs with an often described painful, gripping that can restrict chest expansion. However, SFN is much more extensive and includes autonomic dysfunction, hypohidrosis or hyperhidrosis, gastrointestinal disturbances, e.g., diarrhea, constipation or gastroparesis, micturition disturbances, sicca symptoms (dry eyes and/or dry mouth), blurry vision with accommodation problems, hot flushes, orthostatic dizziness, sexual dysfunction, cardiac palpitations/(pre)syncope [94,97]. SFN occurs as frequently as 86% in northern Europe [23]. The SFN Screening List (SFNSL) can be used to both detect the presence of SFN and track SFN changes overtime in follow-up and management with a minimal important difference of 3.5 point [98].

*Headache* in sarcoidosis requires investigation, and may arise from disease-related events, e.g., CNS, ocular or hormonal complications. Medications can cause general headache symptoms, e.g., glucocorticoids or methotrexate, or poor sleep quality related to glucocorticoids or underlying sleep apnea. Glucocorticoids commonly cause elevated intraocular pressure, beyond normal pressures, and not only cause pain, but can lead to ocular damage. Chronic non-steroidal anti-inflammatory drug (NSAID) use or sensitivity can also produce an aseptic meningitis picture; this may also occur with idiosyncratic reactions to some disease-modifying anti-rheumatic drugs (DMARDS). Importantly, a headache can be a sign of systemic infection.

*Eye pain* in sarcoidosis is often associated with a worrisome disease-related manifestations, e.g., uveitis, glaucoma, optic neuritis, glandular involvement, or medication-related causes. As above for headache, glucocorticoids can increase intraocular pressure creating pain, and both prednisone and biologic agents can increase risk of infection-related ocular complications, such as tuberculosis or zoster. Referred pain or pressure from sinusitis or sarcoidosis sinus involvement can also result in ocular pain.

Facial pain in sarcoidosis can result from multiple causes, including disease- or infection-related trigeminal neuralgia as with zoster or sinusitis; as well as from sarcoidosis-related causes: SFN, cranial nerve, facial bone, or sinus involvement.

Headache and facial pain, especially jaw pain, can also arise from cervical, shoulder, and mandibular muscle tension related to deconditioning, being predominantly sedentary and psychologically high stress levels, which underscores the importance of adhering as much as possible to WHO physical activity recommendations [99].

*Bone pain* in sarcoidosis can be related to osseous sarcoidosis, which, if not treated early, can result in ongoing and more painful destruction and disability. Glucocorticoids are culprits in osteoporotic fractures and not uncommonly avascular necrosis in sarcoidosis patients.

*Joint pain* in sarcoidosis can predominate in the feet and ankles, but is also common throughout other joints in the body and often a sign of systemic disease. Patient reports of joint involvement can be related to osseous sarcoidosis. Again, considering a wide differential including infection, other autoimmune diseases, malignancy, and commonplace causes of joint pain, e.g., gout, etc., is in the best interest of the patient.

*Skin involvement* in sarcoidosis has diffuse presentations and can mimic most other skin conditions. Sarcoidosis-related skin discomfort, in addition to SFN, includes common painful skin manifestations, such as erythema nodosum, panniculitis and, less frequently, painful and/or pruritic cutaneous granulomatous inflammation, and rarely ulcerations. Medication-related skin discomfort, especially with glucocorticoids, includes infections, such as candida, or zoster, skin fragility, tears, and sensitivity.

Other potential pain entities include renal calculi in active, high granuloma volume sarcoidosis phenotypes. As the kidneys strive to prevent calcium homeostasis from tipping into hypercalcemia, upon reaching calcium-secreting capacity, calcium begins to precipitate in the kidneys.

### 5.4. Fatigue

Fatigue is reported as the most frequent and highly impacting symptom affecting patients with sarcoidosis [2,39]. Fatigue is a multi-dimensional, often multifactorial, symptom; often with overlapping causes. Fatigue crudely falls under mental and physical fatigue under which a spectrum of inter-related facets are considered: cognitive, emotional, motivational, physical, muscular, central nervous system, etc. Fatigue is modulated by numerous physiologic and psychological mechanisms directly and indirectly related to sarcoidosis and other diseases: inflammatory/cytokine, mitochondrial, hormonal, hypothalamic, circulatory, neurological, and psychiatric mechanisms, as well as psychosocial influences. Fatigue in other systemic diseases, such as systemic lupus erythematosus and rheumatoid arthritis, are acknowledged to correlate with inflammatory disease activity [100,101]. It is important in sarcoidosis management, to recognize that a myriad of factors drive fatigue and its many components.

#### 5.4.1. Physical Fatigue

Physical fatigue arises from multiple disease-related causes [102]. Patients with pulmonary and extra-pulmonary sarcoidosis report higher fatigue levels than those with solitary pulmonary disease, suggesting an additive impact [103,104]. Systemic inflammatory disease may generate overwhelming fatigue whereby treating the underlying inflammatory condition can provide relief. Suboptimal treatment of sarcoidosis can also result in lingering fatigue, and possibly cognitive dysfunction [94]. Prior active inflammatory disease might result in residual changes to the muscular and vascular infrastructure, potentially resulting in a lower exercise capacity and physical fatigue. Debilitating illness creates a situation whereby, patients increasingly become sedentary and increasingly lose muscle mass and conditioning, leading to weakness and fatigue. For each of these, as discussed below, targeted physical training, can alleviate fatigue. However, decreased muscle endurance or strength can be perceived as ‘fatigue’; patients often do not recognize the presence of sarcoidosis-related or steroid-related myopathy found on PET–CT and/or physical exam.

#### 5.4.2. Other Types and Causes of Fatigue

Pain is a prototypical example of a symptom that can levy both physical and mental types of fatigue as pain impedes physical function and requires mental energy to cope and manage. Sarcoidosis pharmacological treatment can be another source of mental and physical fatigue. Fatigue can also be related to other treatable medication-related side effects, such as nausea, both of which can sometimes be mitigated by adjusting formulation, route, timing, or rate of titration of medication. Other related psychological realms of fatigue can arise from everyday cognitive failure; depressive symptoms, SFN, and dyspnea are identified as positive predictors of fatigue [40]. Anxiety and depressive symptoms potentiate fatigue; chronic fatigue has been successfully treated with cognitive–behavioral therapy.

#### 5.4.3. Assessment and Management Considerations of Fatigue

As above, the FAS, which has both a mental and physical component, is used for both measuring severity and change over time [105], correlates to sarcoid-specific and generic HRQoL measures [106], is available in 21 languages, and can be used in clinical practice and clinical research to quantify and monitor follow-up changes in fatigue [107].

Management of fatigue is essential for enhancing HRQoL in sarcoidosis, and begins with a good history and exclusion of commonplace treatable conditions, such as anemia, coronary artery disease, sleep apnea, etc.; as well as considering sarcoidosis-related culprits, e.g., yet diagnosed pulmonary hypertension, neurosarcoidosis, myopathy, and hormonal dysfunction. Several studies support increasing physical activity and exercise to significantly reduce fatigue and depressive symptoms in sarcoidosis [108,109,110,111], with the wearing of an activity tracker greatly influencing patient increase in physical activity and correlative decrease in fatigue levels [112]. Neurostimulants, such as methylphenidate and modafinil, might be useful in select patients.

### 5.5. Issues of Sleep Quality in Sarcoidosis

Poor sleep quality occurs in as high as 67% of sarcoidosis patients [113] and influences mental health, self-esteem, and inflammation levels [114,115,116,117,118,119]. As with fatigue, potential causes for impaired sleep quality are multiple in sarcoidosis and impaired sleep may be multifactorial [120], with high incidence of obstructive sleep apnea (OSA), restless leg syndrome, and significant periodic leg movement. Sleep quality worsens with increasing dyspnea [113,114], with worse sleep quality associated with high rates of fatigue, depression, anxiety, and cognitive dysfunction. Excessive daytime sleepiness (>10 Epworth Sleepiness Scale) is extremely common in sarcoidosis, and associated with high levels of fatigue, anxiety, and depression, along with diminished physical function and HRQoL [115]. Further, sleep-disordered breathing and OSA have high prevalence in sarcoidosis, and may be worsened by glucocorticoid treatment [116,120].

Sleep quality and inflammation have a bi-directional interplay [117,118]. Poor sleep quality has been inversely associated with inflammation in many health conditions. Cytokine cascades mirroring those found in active sarcoidosis have been shown to be reduced with improved sleep [117,118,119]. The triad of sleep quality, inflammation, and exercise is intriguing. Upon muscle contraction, multiple myokine-driven pathways impact inflammation and CNS mediators of anxiety and depression, as well as promotion of restorative sleep [121,122,123,124,125,126]. Sleep quality may have treatment implications in inflammatory diseases, such as sarcoidosis, with a role for historical assessment and, where applicable, diagnostic follow-up and intervention.

### 5.6. Cardio-Respiratory Symptoms: Breathlessness and Cough

Cardiopulmonary involvement is a predominant feature of sarcoidosis with significant morbidity and mortality related to interstitial lung disease (ILD) bronchiectasis, obstructive, and infectious lung diseases, as well as pulmonary hypertension. Cardiac decompensation may occur secondarily to sarcoidosis pulmonary manifestations, or from primary sarcoidosis-related myocardial or pericardial disease, or neuronal dysfunction. Breathlessness (dyspnea) is common in sarcoidosis, and correlates with exercise intolerance, diminished HRQoL due to physical impairment, and depressive symptoms [113,114,115].

In pulmonary sarcoidosis, a disabling inspiratory cough can exacerbate breathlessness [3,37,127,128,129,130] and by unconsciously restricting or slowing activity levels to avoid symptoms many patients may not recognize mild–moderate symptoms, but know that they are slower and their activity capacity is less. Musculoskeletal impairments, sarcoidosis-related myopathy, and lower extremity joint involvement restrict activity, also making recognition of cardiopulmonary limitations less apparent. However, when queried with contextual and temporal questions, patients might recognize that overtime they have avoided certain activities, e.g., walking up-slope, bending over, or increased the time to walk a distance or make the bed. Understanding and helping patients name/identify components of impairment improve HRQoL by lifting the vague perception that ‘everything’ is helplessly wrong and by providing a definable target/s for improvement.

With cardiopulmonary conditions, *breathlessness* carries neurophysiologic, cognitive and emotional distress, unlike other exertional breathlessness [131,132,133]. Patients worry considerably over breathlessness and what being breathlessness at any given moment might signify. This distress surrounding breathlessness hinders life participation and healthful exercise. Patient–clinician discussions conveying reassurance that *breathlessness* and *desaturation* are distinct attributes, to some degree independent of each other, are key to successful, confident patient engagements [3,38,131,132]. Desaturation is a chemical phenomenon; while breathlessness is a complex multi-factorial, multi-dimensional experience that might include deconditioning, anxiety, unhelpful breath patterns in addition to the cardiopulmonary condition, and of itself, breathlessness is not physiologically harmful [132]. For patients who feel fearful or frustrated regarding physical activity, the following reassurance might be provided: *‘Being physically unfit causes breathlessness and fatigue*’, ‘*Exercise/physical activity helps to increase fitness, exercise also causes breathlessness*, but *can be done in non-distressing manner to help diminish breathlessness over time*’.

Another pervasive opportunity for intervention in cardiopulmonary symptoms relates to dysfunctional breathing patterns, such as hyperventilating or breath-holding. Patients with underlying cardiopulmonary conditions, may be at higher risk for unhelpful habituated breathing patterns that contribute to neurophysiologic mechanisms super-imposed on the breathlessness sensations of the primary health condition [134]. Breathing pattern disorders and dysfunctional breathing patterns can be successfully re-habituated to more supportive respiratory patterns with practiced breath regulation, afforded by physicalizing the breath through exercise and mindfulness techniques, by using body sensation as anchor for breath stability and strengthening. Handheld fans for breathlessness symptoms work in a similar way, with one’s attention being directed to the cool sensation on the skin, which also stimulates parasympathetic activity, slowing heart and respiratory rate.

### 5.7. Exercise Intolerance and Muscle Impairment

Exercise intolerance (45%), as well as muscle weakness (prevalence 12–27%) are common in sarcoidosis [102], with many patients suffering from deconditioning and disability. Exercise intolerance is most commonly considered as rooted in a cardiopulmonary disease or deconditioning. However, overlooking other possible causes, often multi-factorial, diminishes management opportunities. Common contributors of exercise intolerance include myopathy, neuromuscular dysfunction, psychological distress, cognitive disorganization, fatigue, glucocorticoid use, and overt or subclinical inflammation. Evaluation of exercise capacity, as well as muscle function (strength and endurance) assessments, are key assessments for severity and extent of treatable disease in symptomatic sarcoidosis patients and, therefore, HRQoL [108].

In sarcoidosis, patients with impaired peripheral muscle strength are more fatigued and demonstrate impaired pulmonary function testing, six-minute walking distance (6WMD), and HRQoL compared with patients without reduced peripheral muscle strength [102,108]. The relationship between decreased muscle strength, exercise limitation, and fatigue in sarcoidosis is not well-understood, but may be explained by multi-factorial skeletal muscle weakness (granuloma, steroid, systemic inflammation, disuse atrophy) combined with a negative vicious cycle of decreasing physical activity and pulmonary function and increasing deconditioning [102,135,136,137]. ‘Asymptomatic’ myopathy is found in 50–80% of patients with sarcoidosis, whereas clinical symptoms are reported in less than 5%. Myopathic symptoms may be conceivably confused by overlapping fatigue and cardiopulmonary deconditioning. Significant muscle uptake on global PET–CT, suggesting inflammatory myopathic processes, occurs in approximately 12% of patients with sarcoidosis [138], with the remainder of sarcoidosis-myopathy likely being related to non-inflammatory myopathic processes, e.g., steroid myopathy, atrophy, or fibrotic involvement [138].

Pulmonary hypertension (PH) is also under-recognized and, though the true prevalence is unknown, was reported in 6–23% of sarcoidosis patients at rest and in greater than 40% of patients during exercise [139]. The prevalence of PH increases in advanced parenchymal disease portending a significantly worsened prognosis [140]. Underlying, PH may be an occult contributor of exercise intolerance due to both cardiopulmonary and systemic effects of cardiopulmonary compromise on circulation, muscles, and nerves.

Exercise tolerance is often measured by *exercise capacity* or cardiorespiratory endurance or fitness. Muscle is an important organ to sustain this cardiopulmonary fitness as measured by muscle endurance. In clinical practice, several tests are used to evaluate exercise capacity, and can be subdivided in maximal exercise testing (cardiopulmonary exercise test (CPET)) and submaximal exercise testing (six-minute walking test (6MWT)); however, impairment of muscle endurance, rather than momentous exertion of strength, is at the heart of most myopathies, including sarcoidosis-related myopathy. At this time, routine accurate assessments of muscle endurance in sarcoidosis have not been undertaken, but we suggest examining the use of a validated muscle endurance specific assessment, the Functional Index 2 (FI-2) and the FI-3, which examines the endurance of 7 and 3 muscle groups, respectively [141].

The assessment of exercise tolerance, muscle function, physical activity level and exercise engagement, may lead to interventions, such as referral to physical training or exercise counselling that can be pivotal to enhancing HRQoL. Prospective studies are warranted to identify optimal training parameters, duration, frequency, and ways to achieve an optimal long-lasting effect.

## 6. Medication-Related and Complication-Related HRQoL

### 6.1. Medication-Related HRQoL

#### 6.1.1. Adverse Outcomes

Management of medication side effects, offers further opportunity to improve HRQoL. Clinician acknowledgement that, although anti-sarcoidosis medication may effectively treat sarcoidosis and thereby improve HRQoL, the side effects of medication may paradoxically worsen HRQoL. Again, if the clinician focuses solely on objective measures of physiologic improvement and lessening of granulomatous inflammation, medication side effects that impair HRQoL may go undetected.

The impact of glucocorticoids on HRQoL and adverse outcomes cannot be overstressed. While in some cases life-saving, glucocorticoids predictably create short-term intolerability and pose long-term risk of lasting detrimental harm to multiple organ systems. Long-term use of glucocorticoids has resulted in reports of multiple and diverse adverse events in sarcoidosis patients on prednisone and similar glucocorticoids [11,24,142,143,144], including severe infection, psychosis, insomnia, cataracts, glaucoma, gastrointestinal bleeding, myopathy, hormonal dysfunction, and skin and mucosal fragility [145].

One study demonstrated a significantly worse HRQoL for patients receiving even mild doses of prednisone [11]. This includes weight gain, which was found to be dose related [146]. Across >900 sarcoidosis patients from the UK, U.S., and the Netherlands, the strongest association between a reported side effect and drug use was that of weight gain associated with increased appetite among prednisone use [24]. In addition to obesity, glucocorticoids predictably causing hyperglycemia, hypertension, and obesity are particularly worrisome in populations that are at high risk for cardiovascular morbidity [30,31,32,144], which, along with other steroid-related complications, have higher impact by race, gender, and economic status. Avascular necrosis, a source of severe pain, disability, and cause for surgery, and low bone density with a fracture, commonly complicate long-term use of glucocorticoids [30,31,32]. Discussions in the expert community favor elevated BMI, the presence of cardiovascular disease, diabetes, low bone mineral density, prior avascular necrosis, prior gastrointestinal bleeding or ulceration, and psychiatric history as relative contraindications to glucocorticoid use.

Other anti-sarcoidosis medications, if implemented and monitored appropriately, have good safety profiles. The authors employ prednisone judiciously, as dictated by disease severity, and only as a bridge to on-boarding steroid-sparing anti-sarcoidosis treatment, such as methotrexate. This is followed by vigilant proactive tapering of prednisone as the steroid-sparing treatment gains efficacy [30,31,32,147]. Table 2, Table 3 and Table 4 outline common adverse events, contraindications, and protective measures with glucocorticoid use.

#### 6.1.2. Enhancing Treatment Tolerability

Based on an SDM model of communication, employing anticipatory guidance, vigilant history-taking, and monitoring, along with a combination of ameliorating strategies, can increase adherence and reduce aversion to anti-sarcoidosis treatments. More frequent visits, or checking in with patients during the initiation of medication, can intercept problems, early strategies, such as temporary dose reduction, drug holidays, dose timing, dose splitting, formulation changes, or concomitant PRN medications to improve tolerability. For example, most patients tolerate methotrexate very well; however, a few patients experience nausea or fatigue. Taking with food helps to reduce nausea symptoms, and if taken with evening meal; nausea/fatigue may not be noticeable during sleep. Splitting the methotrexate dose over two days, switching from oral to self-injection; PRN anti-emetics or modulating concomitant folic acid dosing all may help to mitigate symptoms. In addition to these, mindfulness techniques—the practice of bringing one’s nonjudgmental attention into the present moment usually by connecting with the body’s sensation of the breath—help to minimize the intensity of patient symptom perceptions [151].

### 6.2. Complication-Related HRQoL

Prevention of complications related to sarcoidosis or its treatment will protect HRQoL status (Table 5). A good proportion of treatment-related complications are mentioned above. Checklists, again, can help navigate patient experience and prevent complications; supplied below are preventive strategies.
Regularly planned testing for medication toxicity monitoring to ensure avoidance of medication interruption and, at the same time, patient safety [8,152].Prescribe sufficient medication, only up until the time for the next toxicity screening test to avoid prolonged use of medication without toxicity check.Monitoring and logging prednisone and NSAID dosages and duration to support proactive tapering or transitioning to more sustainable medication options.Counseling patients on vitamin D regulation in sarcoidosis to prevent hypervitaminosis.Mitigate the risk of serious infection by:○Ensuring vaccinations are up to date;○Considering antibiotic prophylaxis;○Counseling on the best practices, in terms of prevention, e.g., handwashing, masking, etc.Counseling on medication-related red flags for complications and preventive measures for each medication.Counseling on sunscreen use and sun exposure with certain DMARDs and biologic use [152].Supporting exercise as medicine and a stress reduction strategy are expanded upon below.

## 7. HRQoL Self-Management Strategies for Patients and Family Members

As discussed above, in regards to PEMs, patient education about the disease, medications that treat disease, and elements of the healthcare system are essential to motivating engagement, enhancing adherence, and increasing ability to detect and avert potential complications, and therefore protect HRQoL. Making connections with patient organizations, as described above, can provide sustenance to patients and family. Exposure to interventions that, if habituated to some degree into practice, enhance HRQoL through potential amelioration of disease activity, increased fitness, and physical function and enhancing psychological well-being.

### 7.1. Stress Reduction to Enhance HRQoL

Wellness interventions, such as mindfulness [151], yoga, tai chi, and physical activity can help fortify a patient’s ability to cope more easefully with psychological, physical and with logistical healthcare burdens [49,50,51]. The myogenic influence, discussed below in Section 7.2., and other activities that reduce the stress response, induce diffuse physiologic activity, of which vagus nerve stimulation and conditioning is an important component that sets in motion multi-faceted mechanisms within cardiopulmonary, immune system, brain, and central nervous system, and cultivated gastrointestinal responses. Over time, these practices make it easier to engage and prolong this parasympathetic-driven relaxation response. As an example, with reduced stress response, the myocardium relaxes in response to parasympathetic drive, with heart rate slowing that, over time, leads to strengthening heart rate variability. While chronic stress shortens chromosomal telomeres associated malignancy, autoimmunity and fibrosis, integrative interventions halt and appear to increase telomere length [153,154]. The breadth of interventions can be programmatic/system-derived or self-created (e.g., playing musical instrument, painting, or a bedtime decompression ritual), or self-elected (e.g., joining an exercise class or a support group).

### 7.2. Exercise and Physical Activity to Enhance HRQoL

Exercise cultivates a fitness that can offset cardiopulmonary deficiency, thus facilitating greater ease, capacity and reserve for patients to interface with life experience and activities. Thus, a critical component to HRQoL is the clinician supporting, or enlisting other care team members, in safe exercise as a central part of sarcoidosis care. This applies to all levels of ability or disease severity, while taking into consideration possible flares. The focus of counseling is promoting muscular and cardiopulmonary fitness, stimulating disease-modifying mechanisms of exercise, and increasing physical activity and physical capacity. Clinician-initiated discussions on obesity are unlikely to help the patient feel connected with the important message of fitness [155].

Aerobic and muscle-strengthening exercise is recognized across healthy populations and health conditions to significantly improve HRQoL as well as cardiovascular, endothelial, metabolic/glandular, muscle structure and function, lung mechanics, mobility and systemic inflammation, fatigue and depressive symptoms, with overall beneficial effect on a wide spectrum of physiological and psychological attributes. Pulmonary rehabilitation is a feasible, safe, and effective [156,157] strategy for improving breathing, exercise tolerance, fatigue, and cough through education, regardless of underlying diagnosis (e.g., ILD, PH). Exercise safety and safety parameters in sarcoidosis-related cardiopulmonary involvement [157,158] and programmatic considerations, and enhancements in this population, are outlined in Table 6 and Table 7 [159].

Evidence for the role of physical training in sarcoidosis is limited, but promising, and includes improved exercise capacity [110,111,160], muscle strength [110], and HRQoL [160], and decreased fatigue after a 12-week outpatient physical training intervention. Enduring effects on exercise capacity (VO2max and 6MWD) and HRQoL as measured by the St. George’s Respiratory Questionnaire (SGRQ) were maintained at a 6-month follow-up [160].

Exercise being a practiced coordination of movement with breath, increases skilled capacity for complex, weighty, or effort intense activities. Diaphragmatic strengthening, a consequence of many forms of exercise, is another example of exercise’s ability to fortify respiratory dynamics and to improve attributes that support respiratory capacity, e.g., balance, core strength and lower back health [161]. A large muscle, the diaphragm, potentially supports amplification of healthful myogenic pathways related to inflammation, fibrosis, mental health and cardiovascular health. Thus diaphragmatic multi-impact health strategies are increasingly being studied in inflammatory and cardiopulmonary conditions, such as singing for lung health [162,163,164], yogic breathing, tai chi [49,50,51,165], as well as some yoga and dance techniques [166]. These practices cultivate healthier breath patterns and efficiency, rendering an overall contribution to fitness.

## 8. Conclusions

Clinician-initiated influence on HRQoL hinges on patient-centered values. The possibilities are boundless with potentially profound impacts. Patient-centered strategies need not be grand interventions or heroic operational changes to have meaningful impact. Authentic listening and acknowledgement of distress by the clinician enhances the patient’s motivation and empowerment, and guides the clinician to recognize key symptoms—whether they be physical or psychological—and the degree to which they impair HRQoL. Patients are partners in research, policy-making, education, and supporting their own health and the health of other patients—offering any less than this is an outdated paradigm of patient care.

Symptoms are a major force behind impairment and HRQoL. The identification of symptoms is a gateway to revealing multiple opportunities for enhancing HRQoL and other important health outcomes. A clinician cognizant of the broad spectrum of sarcoidosis manifestations and symptoms, as well as their array of potential related causes, is in a better position to carefully deduce and elucidate probable causes. Critical thinking takes time, but helps avoid misdiagnosis, shortens diagnostic delay, protects patients from poor outcomes, and inspires patient confidence in the clinician. All factors that impact HRQoL.

A welcoming, comfortable, safe environment and dedicated patient-centered communication are pivotal to how patients perceive and engage in their care. Tools, such as checklists, SDM, and patient-reported measures, provide multi-purpose support for clinicians to investigate and enhance HRQoL. Beyond simple detection–intervention use, they can each be tools for communication, in regards to changes in symptom distress and disease milestones, disease activity markers to chart patient-reported disease behavior over time, and development of action plans to prevent recurrent complications.

The recognition and prevention of medication-related symptoms and complications for both short-term and long-term health outcomes and unnecessary disability, especially in regard to glucocorticoids, is a major influencer on HRQoL. The sparing and vigilant use of glucocorticoids with the employment of steroid-sparing anti-sarcoidosis treatment alternatives is supported by the authors as a proactive HRQoL measure.

Table 8 offers a preliminary research agenda to investigate or develop a broad range of HRQoL-related correlations, examinations, and interventions. Fatigue, the symptom most frequently identified as disabling, and also the heterogeneous concept of sarcoidosis ‘flares’, require pressing attention to distinguish between cause, complications, type, and safe interventions for recovery. The development of patient reported measures, such as patient engagement measures (PEM), will contribute to the disclosure of specific self-management capabilities, whereby patient education and skills can be strengthened, as an important strategy to enhancing HRQoL. Further, the development of a patient-reported experience measure (PREM) in sarcoidosis is planned and will help assess and guide clinic-based patient-centeredness on local, regional, and national levels. Health disparities continue to be a central unmet need to improve healthcare and survival. Further, the development and testing of communication protocols and formal checklists to support clinical practice in this very complex disease may bridge gaps in multifaceted disparities and elevate care for people living with sarcoidosis.

## Figures and Tables

**Figure 1 diagnostics-11-01089-f001:**
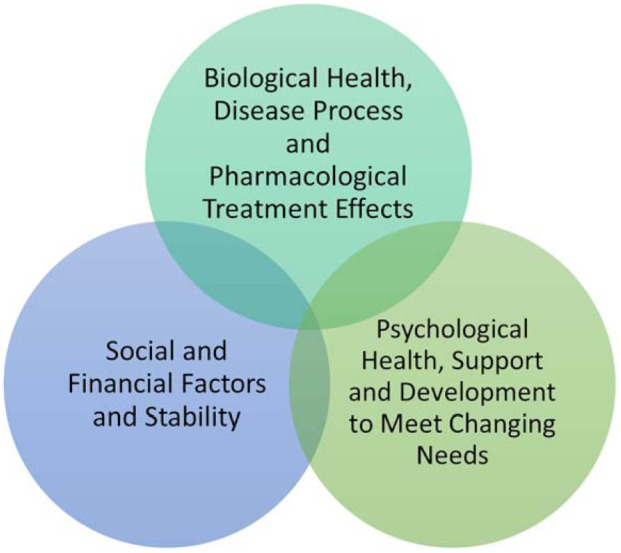
Bio–Psycho–Social Model. The optimization of health outcomes addresses the importance of psychological and social influences on the biological disease process; as well as how the biological disease process impacts essential life areas. (Courtesy of LA Saketkoo, with permission, rights reserved).

**Figure 2 diagnostics-11-01089-f002:**
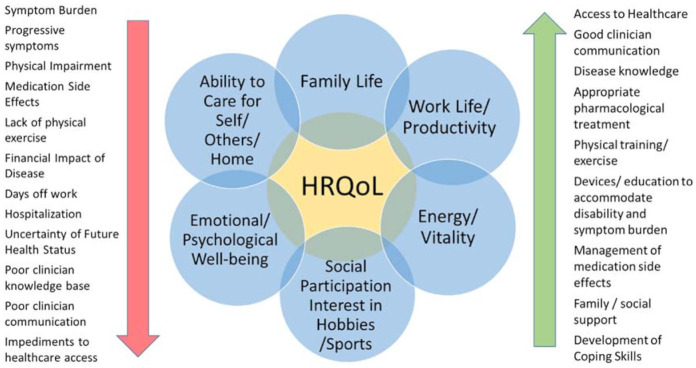
Core components of HRQoL with factors that augment HRQoL on the right or factors that diminish HRQoL on the left (courtesy of LA Saketkoo, with permission, rights reserved).

**Figure 3 diagnostics-11-01089-f003:**
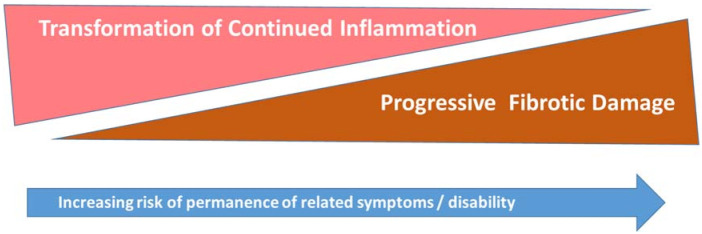
The trajectory of unremitting, ongoing inflammation in relation to tissue damage and reversibility of symptoms. (Courtesy of LA Saketkoo, with permission, rights reserved).

**Figure 4 diagnostics-11-01089-f004:**
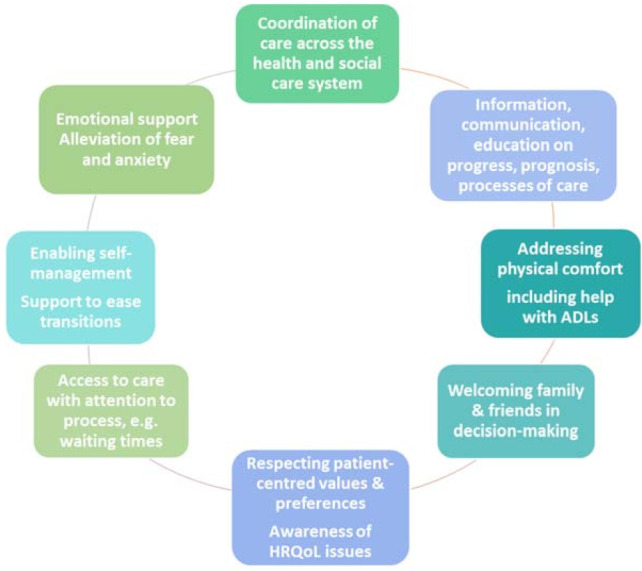
Inter-related principles of patient-centeredness. Modified from Department of Health NHS Patient Experience Framework NHS National Quality Board (NQB) 2011 Gateway reference number 17273 (Public Access Diagram, with unrestricted permission).

**Table 1 diagnostics-11-01089-t001:** Symptom burden is the main driver of diminished HRQoL in sarcoidosis. Symptoms in sarcoidosis can be disease or treatment-related. This table provides a tool to recognize symptoms and initiate reasonable investigations as to cause. Symptoms are often ameliorated with management that either targets the cause or institutes supportive measures that relieves impairment. *Of note, it is important to always consider commonplace causes of symptomatology, in addition to sarcoidosis-related causes.*

Category	Manifestation	Source of Disability/Potential Causes to be Investigated
CONSTITUTIONAL	Fevers	Sub-clinical infection, systemic sarcoid
	Weight Loss	Systemic disease, medication-related nausea, depressive symptoms
	Weight Gain	CNS/endocrine involvement, impairment-related deconditioning, GCs
	Fatigue	Anxiety, body pain, DMARDS/GCs, dyspnea, headache, hormonal (e.g., thyroid, testosterone), GC or anxiety-related insomnia, GC or sarcoid myopathy, HF, hypercalcemia, nutritional, OSA, PH, systemic disease
	Insomnia/Poor sleep	Anxiety, body pain, depressive symptoms, dyspnea, GCs, HF, hypercalcemia, OSA, periodic leg syndrome, restless leg syndrome
PSYCHOLOGICAL	Anxiety	Dyspnea, impaired coping, GC, hypercalcemia, poor sleep quality, uncertainty about future health/finances, vitamin D dysregulation
	Cognitive impairment	Anxiety, body pain, CNS involvement eye discomfort, GC, fatigue, headaches, hypercalcemia
	Depressed feelings	Anxiety, body pain, cutaneous disfigurement; CNS involvement, fatigue, GCs, hypercalcemia, impaired coping, progressive/irreversible impairment, social isolation, uncertainty about future health/finances, vitamin D dysregulation
NEUROLOGICAL	Seizures	CNS involvement
	Headache	CNS, endocrine or ocular involvement, dyspnea, DMARDs/GCs, hypercalcemia, insomnia, OSA
	Weakness	CNS/spinal cord involvement, fatigue, GC or sarcoid myopathy, or large fiber neuropathy, hypercalcemia
	Falls/Gait imbalance	CNS/spinal cord, involvement myopathy, small and/or large fiber neuropathy involvement
	Numbness/Tingling	CNS/PNS, small fiber neuropathy, hypercalcemia
	Dysautonomia (palpitations, sweating abnormalities, orthostatic intolerance, bloating, constipation, diarrhea)	Hypercalcemia, small fiber neuropathy associated symptoms
OCULAR	Acuity impairment	Glaucoma, medication, ocular muscle or nerve involvement, synechiae, uveitis
	Dryness	Lacrimal gland involvement, medication related
	Pain, pressure	Glaucoma, uveitis; CNS, lacrimal gland or sinus involvement,
	Tearing	
OTOLARYNGOLOGICAL	Sinus congestion	GC/DMARD related infection; sarcoid sinus involvement
CARDIAC	Dyspnea, exercise intolerance, palpitations	Conduction abnormalities, HF, hypercalcemia, PH
PULMONARY	Dyspnea, cough, exercise intolerance	Infection, ILD, PH
GASTROINTESTINAL	Dyspepsia	GC/DMARD related
	Nausea, vomiting	CNS, GC/DMARD related, hypercalcemia, renal calculi
ENDOCRINE	Fatigue	Testosterone, thyroid-related, sarcoid or treatment related hyperglycemia, SIADH
	Bone fracture	Systemic disease, GCs, primary or secondary osteoporosis, hormone deficiency
	Weight gain/loss	hypothalamic involvement, inflammation, GC-related
DERMATOLOGICAL	Disfigurement	Dyspigmentation, lupus pernio, facial lesions
	Pruritus	Lesion-related, medication-related
	Pain	Erythema nodosum, ulceration, deep subcutaneous granulomas
MUSCULOSKELETAL		
	Body pain	Boney lesions, joint, hypercalcemia, muscle, neurological, poor sleep
	Bone fracture	See above
	Exercise intolerance	Cardiac or pulmonary involvement (e.g., HF, ILD, PH), Hypercalcemia, Muscle weakness, Deconditioning, Underlying Infection
	Myopathy/Myalgia	Sarcoidosis or GC related
	Weakness	CNS or muscle involvement

Causes of disability and diminished health-related quality of life with preliminary amelioration strategies reported in literature (in sarcoidosis or other conditions) DMARDS: disease modifying anti-rheumatic drugs; GC: glucocorticoids; HF: heart failure; ILD: interstitial lung disease; OSA: obstructive sleep apnea PH: pulmonary hypertension; PNS: peripheral nervous system SIADH: syndrome inappropriate secretion of anti-diuretic hormone.

**Table 2 diagnostics-11-01089-t002:** Adverse side effects of glucocorticoids by system. As glucocorticoids are a major contributor to diminishing HRQoL and long-term complications, recognizing the frequent and common adverse side effects of glucocorticoids is essential to a patient’s health and well-being.

Central Nervous System
Steroid induced psychosis
Changes in mood and behavior
Dysphoria
Insomnia
Changes in memory
Cerebral atrophy
**Ocular**
Glaucoma
Cataracts
**Cardiovascular**
Hypertension
Dyslipidemia
Thrombosis
Vascular frailty
**Gastrointestinal**
Peptic ulcer
Gastrointestinal bleeding
Pancreatitis
Hepatic steatosis
**Renal**
Increased sodium retention
Increased potassium excretion
**Endocrine**
Diabetes mellitus
Weight gain
Cushing’s syndrome
Adrenal suppression
Hypogonadism
Male gynecomastia
**Musculoskeletal**
Osteoporosis
Bone necrosis
Muscle atrophy
Myopathy
**Skin**
Striae rubrae distensae
Skin frailty and tearing
Delayed wound healing
Glucocorticoid induced acne
Secondary infections, including candidiasis
Perioral dermatitis
Telangiectasia
Skin atrophy
Seborrheic dermatitis
**Immune**
Reactivation of latent viruses
Increased risk of infection
Immunosuppression
Serious Infection
Candidiasis: oral/cutaneous/vaginal

**Table 3 diagnostics-11-01089-t003:** Glucocorticoids are associated with a high level of adverse effects, long-term complications, and disability in sarcoidosis. Thus, inappropriately used and monitored, glucocorticoids can result in greater diminishment of HRQoL, the sarcoidosis disease process itself. This table provides a checklist of preventive measures when using glucocorticoids.

Bone Health [148,149,150]
Counsel patients about the risk of osteoporosis and screen for risk factors.
Baseline DEXA scan (for patients anticipated to need glucocorticoids for >3 months). Initiation of bisphosphonate for prevention according to American College of Rheumatology guidelines.
Calcium supplementation is controversial in sarcoidosis and vitamin D supplementation, only if 1,25 dihydroxy is low.
Counseling on lifestyle modifications—smoking cessation, weight-bearing activities.
Baseline height as surrogate for vertebral height/compression fracture.
Gastrointestinal [148]
Counsel on gastric protection, take with food, H2 Blocker, or PPI depending on risk level. Assess for risk factors for PUD–history of PUD, heavy smokers, heavy alcohol use, age >65 years old, other medications that increase risk of PUD.
For patients on glucocorticoids and nonsteroidal anti-inflammatory drugs, start PPI.
For patients with multiple risk factors for PUD, consider addition of PPI.
Endocrinology [148]
In patients with diabetes, glucose monitoring with sliding scale insulin instructions. Consider screening for diabetes–hemoglobin A1C, basic metabolic panel, or fingerstick glucose.
Monitoring fingerstick glucose or basic metabolic panel in patients.
Consider prescribing home glucometer for patients on long-term high dose glucocorticoids.
Monitoring of electrolytes.
Cardiovascular [148]
Baseline lipid panel.
Blood pressure monitoring and treatment of hypertension if indicated.
Immunizations [148]
Inquire about vaccination history.
Live vaccines should be given 2–4 weeks prior to initiation of glucocorticoids if possible.
Administer vaccines according to standard schedule as indicated; withholding live vaccines.
Psychiatric [148]
Inquire about history of neuropsychiatric disease, suicidal ideation, and self-harm.
Referral to psychiatrist if indicated.
Counsel family members on risk of mood and behavior changes and advise physician if any changes are noted.
Dose glucocorticoids in the morning to reduce insomnia. Monitor for insomnia, manage insomnia as needed.
Ocular [148]
Assess for personal and/or family history of glaucoma or cataracts.
Obtain baseline ophthalmologic exam for patients who may need long-term glucocorticoid treatment.
Infectious [148]
Consider PCP prophylaxis for patients taking the equivalent of ≥20 mg prednisone for ≥4 weeks, especially if a second risk factor is present—hematologic malignancy, interstitial lung disease, or use of other immunosuppressant medication.
Inquire about infection history and risk factors for bacterial, fungal, and viral infections, and screen if indicated.

DEXA: dual energy x-ray absorptiometry; PCP: pneumocystis pneumonia; PPI: proton pump inhibitor; PUD: peptic ulcer disease.

**Table 4 diagnostics-11-01089-t004:** This table highlights the relative contraindications of glucocorticoid use that frequently result in poor health outcomes and diminished HRQoL for patients. Though limiting glucocorticoid use for all patients is important, the below conditions are critical considerations for treatment alternatives for glucocorticoids. Or if disease severity (e.g., ocular, cardiac, neuro, etc.) necessitates glucocorticoid treatment, close observation with dose reduction, as soon as alternative treatments take effect, is recommended.

Hypersensitivity to any component of formulation
Concurrent administration of live or live-attenuated vaccines
Depression, uncontrolled anxiety, or history of psychosis
History of peptic ulcer disease or gastrointestinal bleed
Osteoporosis
Current or recent joint infection
Glaucoma/elevated intraocular pressure
Cataract
Diabetes mellitus/uncontrolled hyperglycemia
Uncontrolled hypertension
Elevated BMI/metabolic syndrome
Uncontrolled bacterial or viral infection
Systemic fungal infection

**Table 5 diagnostics-11-01089-t005:** Opportunities for enhancing HRQoL.

Strategies to Improve and Preserve HRQoL
Screening with review of systems; investigate and address other potential sarcoidosis manifestations.
Continually keep in mind that new or worsening symptoms may not be sarcoidosis.
Screening for treatment side effects to improve adherence and tolerability.
Screening for fatigue, assessment, and intervention for predominant causes.
Screening for impaired sleep quality and OSA.
Screening for and treating depression and anxiety.
Ensuring patient health literacy as a priority of treatment.
Essential clinician communication using shared-decision making regarding their interpretation of test results, disease severity, anticipated medication response, and prognosis.
Ascertain, through reflecting back to the patient’s understanding and opinion.
Recognition that disease activity may not be reflected in the results of the ‘objective’ tests and that lack of evidence on ‘objective’ testing does not preclude disease activity.
Anticipatory guidance regarding medication side effects.
Encouragement to perceive exercise as medicine, regardless of ability.
Referral for physical training and pulmonary rehabilitation for those with reduced exercise tolerance and unexplained fatigue. Patient engagement depends on their report of debilitation at any given time, as certain types of flares are globally incapacitating.
Publicize patient support group meetings as well as reliable patient education sources.
Preventive strategies for complications of disease or treatment: immunizations; medication monitoring, bone, and gastric protection; prophylactic antibiotic, as appropriate.
Opportunities for patients to learn skills for well-being, such as coping, mindfulness techniques, and organizational skills.

**Table 6 diagnostics-11-01089-t006:** Stratifies patients with sarcoidosis for engagement in exercise.

Strata of Engagement	Advisement	Comments
**No cardiopulmonary involvement**	Unrestricted but targeted to and guided by patient needs, and tolerance	Gradual increment of intensity, repetitions, and duration
**Mild cardiopulmonary symptoms**	Moderate aerobic intensity with moderate-load resistance exercises	Gradual increment of intensity, repetitions, and duration
**Severe cardiopulmonary symptoms**	Individualized modification of intensity and duration, with supplemental oxygen as needed	Can be intensified up to 75–80% of a patient’s projected maximal load
**Desaturation with exercise**	As above for severe	
**Increase of systolic pulmonary artery pressure with exercise**	Load reduction on systemic and pulmonary circulation is an important consideration	Rapid changes in pulmonary hemodynamics interval training may increase risk of syncope

**Table 7 diagnostics-11-01089-t007:** General considerations for exercise in sarcoidosis.

Concept	Advisement	Comments
**Exercise Initiation**	All patients screened for clinically significant ILD and PH	
	Assess current activity levels with FITT	FITT = Frequency, Intensity, Type, Time, an exercise program/prescription created by patient or clinician
	Consider assessing patient goals with PSFS	PSFS = Patient Specific Functional Scale, a patient created scale of their unique priorities
	Given the fairly high prevalence of sarcoid myopathy and neuropathy, aerobic and muscle testing prior start of exercise	Submaximal ergometer cycle test or treadmill test and muscle tests like TST, 30-sec CST and FI-2
**Sustaining exercise**	Anticipatory guidance of fluctuating fatigue/pain challenging exercise	Encourage mindfulness practice and pleasure principals during exercise to redirect frustration and disappointments
	Education on stretching safety	Emphasis on consistency of practice and expectations of incremental improvement
	Developing alternate options for inclement weather or GI exacerbations	Indoor options Online class options
	Consider monitoring achievement with PSFS	
	Start gently and escalate with improvement	
	Recommendation for home general physical activity (e.g., walks) 30 min/5 days weekly	
	Aerobic and muscle testing after exercise period to evaluate intervention	Submaximal ergometer cycle test or treadmill test and muscle tests like TST, 30-sec CST and FI-2

CST: Chair to Stand Test; FI-2: Functional Index-2; FITT: Frequency, Intensity, Type, and Time (referring to an exercise program); GI: gastrointestinal; PSFS: Patient-Specific Functional Scale; TST: Timed Stands Test.

**Table 8 diagnostics-11-01089-t008:** Research Agenda for Sarcoidosis-Related HRQoL.

Expansive patient perspective investigations.
Isolate determinants of racial disparities in sarcoidosis, develop proactive interventions for targeted reversal of identified disparities.
Characterize presentations and descriptors to help identify type and cause of fatigue in sarcoidosis.
Develop protocol for management of fatigue according to type and cause.
Investigations to characterize ‘flare’ types, relational causes and recovery.
Integrated mind–body strategy impacts on inflammation, pulmonary functioning, fatigue, and HRQoL.
Exercise and physical activity impact on inflammation and other biomarkers, HRQoL.
Testing muscle endurance with FI-2/FI-3 in correlation with fatigue and other assessment parameters.
Identify optimal parameters of physical training in sarcoidosis.
Impact of sarcoidosis-related psychological distress on patient perceptions of their loved ones’ anxiety and emotional distress.
Impact of home-based prescriptions for physical activity and exercise on depressive symptoms, fatigue, inflammation, and activity levels in patients with sarcoidosis.
Examine HRQoL in the context of morbidity and survival in sarcoidosis.
Examine HRQoL interventions on symptom distress and survival.
Examine exercise on HRQoL and survival as stratified for disease severity.
Development of sarcoidosis-specific patient-reported experience measure (PREM).
Development of sarcoidosis-specific patient engagement measure (PEM).
Develop and test CME practice modules on: - Healthcare disparities in sarcoidosis, impact of differential treatment on complications, and financial distress, - HRQoL determinants in sarcoidosis, especially fatigue and medication-related management,

## Data Availability

Not applicable.

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
