# Peer review of "Health-Related Quality of Life (HRQoL) in Sarcoidosis: Diagnosis, Management, and Health Outcomes"

_diagnostics, 2021, doi:10.3390/diagnostics11061089_

Round 1
Reviewer 1 Report
The authors provide tools to assess, investigate, and diagnose compromised HRQoL and its influencers of the sarcoidosis. Strategies to address modifiable HRQoL factors through palliation of symptoms and methods to improve the sarcoidosis health profile are outlined; as well as a proposed research agenda in sarcoidosis-related HRQoL.
The manuscript is well written and many important findings are described. I have some comments.
1. There are no scoring system related to HRQoL of the sarcoidosis. Thus, in Figure 2, It is not clear whether these factors actually improve or exacerbate HRQoL.
2. There is no specific data on the actual patient's HRQoL in this paper.
3. Therefore, the usefulness of HRQoL in patients with sarcoidosis, which we emphasize, is not clear.
Author Response
Dear Reviewer #1,
Thank you for your thoughtful need for clarifications. Your comments are responded to with the corresponding number as you have numbered it.
- Thank you for this. We can see how this might require clarification. We have added the following phrase to the caption for clarification: with Factors that Augment HRQoL on the Right or Factors that Diminish HRQoL on the Left
- Thank you. That is correct. This is a review that synthesizes the diverse and expansive data on the subject of HRQoL in sarcoidosis. There is however, a panel of authors on this paper that are patients who are recognised peer leaders in sarcoidosis, who have guided this document with the priorities they believe to be important to patients and overlooked by clinicians.
- Thank you. This is an astute point, and why we have included the research agenda. There are only two interventions that have been measured to impact some aspects of HRQoL, the small study on mindfulness, and a few studies on exercise.
As you rightly, allude, this is scratching the surface and we need more studies that demonstrate the impact of interventions in HRQoL in sarcoidosis. The trouble is in HRQoL, we have to begin by summarising and quantifying observational data, which is done in this review. At this point, we can begin to launch investigations similar to those done in cancer, cardiovascular disease etc. This is our hope, that this paper will inspire other researchers to take an interest and notice the wealth of observational data collected that demonstrate need.
Reviewer 2 Report
The work titled "Health Related Quality of Life (HRQoL) in Sarcoidosis: Diagnosis, Management and Health Outcomes" deals with an interesting topic. HRQoL may be the outcome most reflective of patient priorities when living with a health condition. HRQoL in sarcoidosis is an expansive topic and the authors want to offer an abbreviated "reference manual" to identify HRQoL aspects that are susceptible to intervention, develop an approach to assessing HRQoL, and consider a preliminary research agenda in sarcoidosis-related HRQoL.
The authors may consider the following comments:
Line 202: Table A is indicated but it is not present in the manuscript. Please resolve this inconsistency.
Line 524: please, specify the meaning of the abbreviation ILD.
Line 530: please, specify the meaning of the abbreviation MCID.
Line 538: please, specify the meaning of the abbreviation MACTAR.
Line 682: please, correct "use d" in "used".
Table 1: In the line that refers to "Exercise intolerance" is missing the description of the "Source of disability/Potential causes to be investigated". Please enter missing description. Also add the meaning of the abbreviations PNS below the table.
Table 2: In the first line replace the abbreviation CNS with central nervous system.
Table 3: please, specify the meaning of the abbreviations PUD, PPI and PCP.
Line 1033: please, specify the meaning of the abbreviation SGRQ.
Table 7: please, specify the meaning of the abbreviations TST, CST and GI.
Author Response
Dear Reviewer #2,
Thank you very much for your care in identifying these. This is very helpful.
Each of these have been located in the manuscript and corrected. The revisions are each found in the manuscript with ‘tracked changes’.
We very much appreciate your time.
Reviewer 3 Report
The review extensively addresses issues related to improving the quality of life of patients with sarcoidosis. The work is very thorough and may be useful to non-experts to punctuate the issue entirely. For experts, it is useful to get an overview of the patiens’problems.
The paper is very long and complex to read.
I personally think that some parts should be streamlined to make it easier to read.
I suggest in particular to :
Chapter 2: to be streamlined, in particular Fig 1 weighs down the text and is not particularly informative. I would remove it.
In line 202 table A seems to refer to Table 1. I would put it in a supplementary file.
The content in the text would be more useful to put using the categories in Figure 2. It is hard to follow the text using different terms and patterns.
Chatpter 3 . Patient centredness. If we use Figure 2 as a conceptual template, how does this part connect with the template?
The text of lines 319-331 is overlapping with Figure 4. I would remove the figure and keep only the text. the sub-points in chapter 3 answer only some of the 8 aspects. why?
Table 2 describes the general side effects of corticosteroids. What does it add specifically? I would remove it or put it in the supplemental files. Same with table 3
Tables 6,7,8 are also insights that weigh down the text. I suggest to put them in the supplementary files.
I propose a reflection to lighten the reading: what is really related to sarcoidosis and what is common as an approach to all chronic maladies? I would keep only what answers the first question.
266 citations are really too many. What is recent and closely related?
Author Response
Dear Reviewer #3,
Thank you very much for these considerate recommendations. We will respond in order of your commentary.
GENERAL: It is understood that the text, albeit very informative, is long; and for some to read through from beginning to end can be tedious. However, the intention of this publication is for the reader to take as little or as much from it as they like, to take an overview or a deep dive. But the main utility is that this publication can be read, re-read, referred to in the middle of clinic, especially for clinicians overwhelmed with the complexity of sarcoidosis. We did try to preface this in the abstract and introduction. I would like to share that we asked other physicians for a ruthless review, as we recognized the somewhat atypical nature born of an urgency to educate and skill clinicians – who in our experience are wanting such. This helped us with some details but the consistent response was that this was a tremendous work that will make a difference for patients and clinicians alike. We hope you might re-read, and re-consider from this perspective.
We have reduced the size of figure 1; and request that the editorial team please feel free to scale as is deemed appropriate. This is for any figure, as we are not used to this format and simply pasted the figures as is. Our reason to advocate keeping this figure is that it provides an imprint on the mind of the reader upon which incremental concepts are then built. Figure 1 is foundational to understanding that health is an intersection of biological and other factors. Even 40 years onward, unfortunately, a large proportion of clinicians are uninitiated to this concept; and can be a transformational revelation especially for procedure-based clinicians.
Thank you. Table A is now corrected to Table 1. We did originally consider putting this table in a supplementary file; however this table representing the diverse, repeated misdiagnosis of complications was a, if not ‘the’, major area of concern as expressed by our patient peer leader panel of authors. These authors not only represent themselves but as national leaders have supported tens if not hundreds of sarcoidosis patients navigate their healthcare needs.
Thank you. Figure 2’s caption has been made clearer to better guide the reader. Again the figure can be scaled down as deemed fit. The terms used in the central concept venn and the outer factors are fairly closely termed in the text; one exception might be referring to fatigue rather than vitality/energy; however vitality /energy is the more positive term that is aspired to, and this is in keeping with the rest of the concepts stated the positive term of the concept. We felt this is an important impression/imprint to provide the reader, whether they are patients or clinicians; it is the aspiration toward improvement.
Thank you. As asserted throughout the chapter and dissected in the first two paragraphs of the section (hopefully the pasted below provides sufficient support to your query), it is essential. Patient-centredness is the seed and the engine required to accurately and meaningfully enhance HRQoL through both clinical care and research pathways:
Clinician-initiated strategies to increase HRQoL, begin with an environment whereby a patient feels respected, listened to and believed, followed by open-minded investigations based on patient history. Subsequent efforts target the patient’s report of symptom burden, physical impairment, as well as potential adaptations to home/work participation. This occurs in the context of continued thoughtful consideration of resources at hand, such as other available programs and inter-disciplinary care members such as occupational or physical therapy.
Patient-centeredness is at the heart of approaches and actions directed toward increasing a patient’s HRQoL within the sphere of healthcare delivery. Patient-centeredness broadly translates to patients’ voiced priorities, concerns, and perspectives that can guide anything to do with care on large scale or individual levels. This includes treatment/management, communication, clinical environment, policy and procedures, in addition to delivery of and access to care. Figure 2 outlines the powerful impact that clinician-related factors have on HRQoL.
Thank you. Figure 4 addresses other concepts included elsewhere in the paper. The figure, we agree is not absolutely necessary; but we thought provides a visual of these central concepts which we thought might serve some of the more visual readers. If after our explanation, you feel the figure truly does not serve the text, we agree for the editors to remove it.
Tables 2 and 3, thank you for the opportunity to discuss these tables. The misuse and the mindless use of glucocorticoids in sarcoidosis is a large topic with multiple aspects that we feel warrant itemizing in table and explaining in the test. Again, these tables are related to. Glucocorticoids are central to patient concerns and where they see continued lack of cognizant use by clinician which results in continued long and short-term suffering and unnecessary suffering and disability. As clinicians of sarcoidosis centers inheriting patients with an established sarcoidosis diagnosis, we can each report the high volume of unchecked use of prolonged glucocorticoid use, the lack of preventive adjuvant care and the harm that this has caused. Thus we feel it imperative that these tables remain central to the text. We hope after reading our explanation that you will agree.
Thank you, for reasons stated above, we would humbly request that tables 6, 7, 8 remain part of the main text. These tables are meant to be used for reference and cross-reference in a clinic to help trouble shoot causes and treatment of symptoms and complications of this very complex illness.
HRQoL in sarcoidosis is an expansive, diverse topic. The reference list is extensive which consequently demonstrates what is known but despite how much is known, it also lay down many pathways for future research and testing of diverse HRQoL interventions. We used the most up to date sarcoidosis-specific articles wherever we could to substantiate our points; however we used some older articles that were foundational or sentinel to certain concepts. We hope the reviewer had an opportunity to read through the references to note the diversity and novelty of the titles between each other and their specificity to sarcoidosis; and where not sarcoidosis-specific per se, the reference was used to substantiate a point that did not yet have a parallel reference directly for sarcoidosis.
Thank you very much for this careful, thoughtful review and the opportunity for us to re-review and re-consider points of the article.
Round 2
Reviewer 3 Report
I note that the authors have decided to ignore the suggestions made. Good luck with the article.
Best regards
Author Response
Dear Reviewer #3,
We hope that you were able to see our thorough replies to each of the comments - reviewer #3 helped significantly in illuminating where further attention was required.
None of the reviewers' commentary would have been intentionally dismissed. We provided what we believed to be thorough replies after careful consideration for each point that was supplied by each of the authors.
Our language in the responses was intended to be an explanation as to why we felt strongly about something, but also how we would modify to address whatever lack in our writing impaired a reader's ability to see the connection between the essential points we intended to make.
We intended that our writing in an ongoing fashion, sought reviewers' approval for changes.
It is our responsibility as authors to improve the writing based on reviews - and the writing of the manuscript from our point of view has certainly improved because of reviewer commentary - but it is also our responsibility to explain what was intended. We hope that it is the collaborative effort between writers and reviewers, that the integrity of the manuscript is furthered.
We very much valued the commentary by reviewer #3, and the time dedicated to provide the helpful review, there was no commentary that we did not take seriously. It is very much hoped, that reviewer #3 can see the value placed on their comments by our thorough responses.
Best wishes -
Lesley Ann Saketkoo on behalf of the author team.